# Comprehensive analysis of full-length transcripts reveals novel splicing abnormalities and oncogenic transcripts in liver cancer

**Hiroki Kiyose[1,2], Hidewaki Nakagawa[3], Atsushi Ono[4], Hiroshi Aikata[4], Masaki Ueno[5], Shinya Hayami[5], Hiroki Yamaue[5], Kazuaki Chayama[6,7,8], Mihoko Shimada[1], Jing Hao Wong[1], Akihiro Fujimoto[1]** *

1 Department of Human Genetics, Graduate School of Medicine, The University of Tokyo, Tokyo, Japan,
2 Department of Drug Discovery Medicine, Kyoto University Graduate School of Medicine, Kyoto, Japan,
3 Laboratory for Cancer Genomics, RIKEN Center for Integrative Medical Sciences, Yokohama, Japan,
4 Department of Gastroenterology and Metabolism, Graduate School of Biomedical and Health Sciences, Hiroshima University, Hiroshima, Japan, 5 Department of Surgery II, Wakayama Medical University, Wakayama, Japan, 6 Collaborative Research Laboratory of Medical Innovation, Graduate School of Biomedical and Health Sciences, Hiroshima University, Hiroshima, Japan, 7 Research Center for Hepatology and Gastroenterology, Graduate School of Biomedical and Health Sciences, Hiroshima University, Hiroshima, Japan, 8 RIKEN Center for Integrative Medical Sciences, Yokohama, Japan

* afujimoto@m.u-tokyo.ac.jp

## Abstract

Genes generate transcripts of various functions by alternative splicing. However, in most transcriptome studies, short-reads sequencing technologies (next-generation sequencers) have been used, leaving full-length transcripts unobserved directly. Although long-reads sequencing technologies would enable the sequencing of full-length transcripts, the data analysis is difficult. In this study, we developed an analysis pipeline named SPLICE and analyzed cDNA sequences from 42 pairs of hepatocellular carcinoma (HCC) and matched non-cancerous livers with an Oxford Nanopore sequencer. Our analysis detected 46,663 transcripts from the protein-coding genes in the HCCs and the matched non-cancerous livers, of which 5,366 (11.5%) were novel. A comparison of expression levels identified 9,933 differentially expressed transcripts (DETs) in 4,744 genes. Interestingly, 746 genes with DETs, including the *LINE1-MET* transcript, were not found by a gene-level analysis. We also found that fusion transcripts of transposable elements and hepatitis B virus (HBV) were overexpressed in HCCs. *In vitro* experiments on DETs showed that *LINE1-MET* and HBV-human transposable elements promoted cell growth. Furthermore, fusion gene detection showed novel recurrent fusion events that were not detected in the short-reads. These results suggest the efficiency of full-length transcriptome studies and the importance of splicing variants in carcinogenesis.

## Author summary

Genes generate transcripts of various functions by alternative splicing. However, in most transcriptome studies, short-reads sequencing technologies (next-generation sequencers)

**Data Availability Statement:** The source code of SPLICE is freely available from Github (https://github.com/hkiyose/SPLICE). The sequencing data have been deposited in the National Bioscience

Database Center (NBDC) under accession JGAD000635 (https://humandbs.biosciencedbc.jp/en/hum0182-v5). The sequencing data is controlled access, and an application to the NBDC is needed.

**Funding:** This work was supported by grant 18H02680 from the Japan Society for the Promotion of Science (JSPS) to AF, and by grant JP21km0908001 from Japan Agency for Medical Research and Development (AMED) to AF, and by grant from Takeda Science Foundation to AF. The funders had no role in study design, data collection, and analysis, decision to publish, or preparation of the manuscript.

**Competing interests:** The authors have declared that no competing interests exist.

have been used, leaving full-length transcripts unobserved directly. In this study, we developed an analysis pipeline named SPLICE for long-read transcriptome sequencing and analyzed cDNA sequences from 42 pairs of hepatocellular carcinoma (HCC), and matched non-cancerous livers with an Oxford Nanopore sequencer. Our analysis detected 5,366 novel transcripts and 9,933 differentially expressed transcripts in 4,744 genes between HCCs and non-cancerous livers. An analysis of hepatitis B virus (HBV) transcripts showed that fusion transcripts of the HBV gene and human transposable elements were overexpressed in HBV-infected HCCs. We also identified fusion genes that were not found in the short-reads. These results suggest that long-reads sequencing technologies provide a fuller understanding of cancer transcripts and that our method contributes to the analysis of transcriptome sequences by such technologies.

## Introduction

Hepatocellular carcinoma (HCC) is the third-leading cause of death worldwide and the seventh most common form of cancer [1]. Common etiological factors in liver carcinogenesis include infection by hepatitis B virus (HBV) or hepatitis C virus (HCV), but other factors, such as alcohol intake, metabolic diseases, and exposure to specific carcinogens also play significant roles [2]. These factors cause liver inflammation, leading to cirrhosis and result in malignant transformations in hepatocytes [2]. To elucidate the molecular mechanisms underlying liver carcinogenesis, genetic and transcriptional aberrations have been investigated [3–7]. These previous studies identified numerous somatic mutations and differentially expressed genes (DEGs), and have led to the discovery of liver cancer-associated pathways, such as apoptosis, Wnt signaling, chromatin remodeling, and lengthening telomeres [3–7]. In HBV-infected liver cancers, HBV integrations and HBV-human fusion transcripts have been identified [8]. Fusion genes have been investigated, but recurrent fusions were rare [9]. Although these studies have expanded our knowledge of HCCs, our understanding of liver carcinogenesis is far from complete [10,11]. To obtain deeper insight into the mechanism responsible for HCCs, applications of novel technologies and analysis of new aspects of genetic and transcriptional aberrations are needed.

We aimed to analyze full-length transcripts for HCCs in the present study. The majority of previous transcriptome studies have used microarrays or short-reads sequencing technologies, and therefore lacked information on full-length transcripts that may assist in the detection of splicing variants expressed from each gene. Protein sequences are specified by the mRNA sequences of transcripts; thus, the direct observation of transcripts should provide essential information about carcinogenesis. Indeed, several recent studies have suggested that transcript-specific functions contribute to carcinogenesis in breast and ovarian cancers [12–14]. In HCCs, a recent study analyzed short-reads RNA-seq data and detected oncogenic splicing changes in the *AFMID* gene [15]. These splicing changes were estimated to occur at the early stage of HCC carcinogenesis and were associated with patient survival [15]. Overall, previous studies strongly suggest that splicing variants have important roles that are not recognized without an analysis of full-length transcripts.

One promising approach to observe transcripts is RNA-seq using long-reads sequencers [16], but their application in cancer research is still limited [17–21] for reasons such as high error rates in the long-reads sequencing [22] and the complexity or heterogeneity of the cancer genome. To overcome this problem and identify cancer-related transcriptional changes, we developed a method for analyzing RNA-seq data obtained by a long-reads sequencer, MinION

(Oxford Nanopore). In this study, we sequenced 42 pairs of cDNA from HCCs and matched non-cancerous livers that were previously used for whole-genome sequencing (WGS) and RNA-seq of short-reads [4,23,24]. We developed an analysis method named SPLICE and analyzed the long-reads data. Among the reads mapped to coding genes, 61.7% contained entire coding sequences, and a total of 46,663 transcripts were detected in the protein-coding genes in the HCCs and matched livers, of which 5,366 (11.5%) were novel. A comparison of expression levels found 9,933 differentially expressed transcripts (DETs) in 4,744 genes. Interestingly, 746 genes with DET were not detected by the gene-level analysis. Exonization of transposable elements (TEs) was also identified. In the analysis of HBV transcripts, fusion transcripts of the HBV HBx gene and human TEs were overexpressed in HBV-infected HCCs. These results suggest that long-reads sequencing technologies provide a fuller understanding of cancer transcripts and that the SPLICE method contributes to the analysis of transcriptome sequences by such technologies.

## Results

### cDNA sequencing with Oxford Nanopore sequencer

We sequenced cDNA from MCF-7 cells and previously published 42 cancer and their matched non-cancerous liver pairs (**S1 and S2 Tables and S1 and S2 Data**) [4]. cDNA was generated by reverse transcription with a SMARTer PCR cDNA synthesis kit (TakaraBio), and sequencing libraries were constructed using the SQK-LSK109 library construction kit (Oxford Nanopore) according to the manufacturer's instruction. We performed one run for each sample using flowcell R9.4 (Oxford Nanopore). The data yield ranged from 4.6 Gbp to 20.6 Gbp (average 12.7 Gbp), and the number of reads was 3,061,266 to 14,873,634 (average 9.1 M reads) (**S2 Data**).

### Construction of analysis pipeline and evaluation of the SPLICE method

To analyze the long-reads data, we constructed an analysis pipeline named SPLICE (see **Methods**) (**Fig 1A**). Long-reads sequencers have the capability to sequence entire cDNA end-to-end [25], allowing us to capture full-length transcripts without assembly [26]. However, long-reads sequencers also have high error rates [22], which could lead to incorrect conclusions about splicing variants and their expression levels. To overcome these obstacles, we analyzed error patterns and developed an analysis pipeline.

Reads were mapped to the reference genome sequence (hg38), reference transcriptome sequence (GENCODE (version 28), and RefSeq (release 88)) using minimap2 software [27]. We found that false-positive splicing aberrations were caused by alignment errors at splicing junction sites, mapping errors due to highly homologous genomic regions, and artificial chimeric reads (**Fig 1B**). The following describes how the SPLICE pipeline removed possible error patterns from the mapping results (**S1 Fig**). (1) Reads were mapped to the reference genome and the reference transcriptome sequences, and reads were removed if the number of matched bases in a read to the reference genome sequence was smaller than that to the reference transcriptome sequence (removal of mapping errors). (2) For the identification of novel transcripts (novel splicing variants), we evaluated the error rate within ± 5-bp regions from splicing junction sites and removed candidates for error rates ≧ 20% (removal of alignment errors around splicing sites). (3) Because fusion gene candidates can be generated by mis-ligation of two transcripts during the library preparation, we removed fusion gene candidates that contained primer sequences for cDNA synthesis between two genes (removal of artificial chimeras). (4) Mapping errors due to highly homologous genomic regions also can cause fusion gene candidates; thus, we compared the results of mapping to the reference genome and the

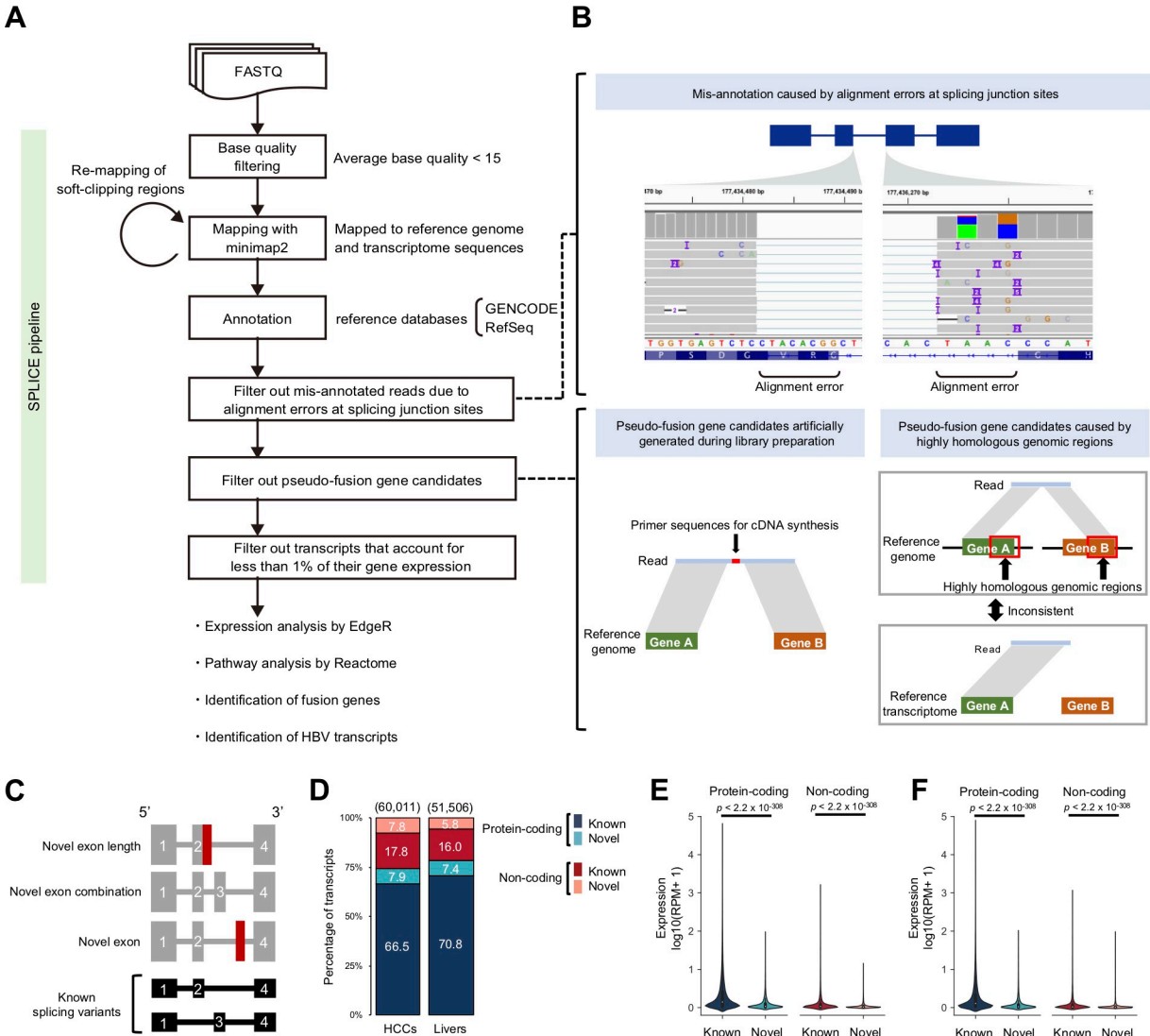

**Fig 1. Overview of the SPLICE pipeline and HCC sequencing library.** (**A**) Workflow of the SPLICE pipeline and data analysis of this study. Reads with an average base quality score < 15 were removed from the input FASTQ files. The read sequences were mapped to the reference genome, reference transcriptome sequences, and remapped soft-clipped regions (≧ 60 bp). Candidates corresponding to the error patterns or low-frequency transcripts were removed. The final data were used for the subsequent data analyses. (**B**) Error patterns in cDNA-seq using MinION. Alignment errors caused by sequencing errors, artificial chimeras, and mapping errors of highly homologous genes can result in errors. (**C**) Classification of patterns of novel transcripts. "Novel exon length" contains known exons with different lengths from the reference database. "Novel exon combination" has a different exon combination from the reference database. "Novel exon" contains exons expressed from regions that are not overlapped with known exons. (**D**) Composition of the transcripts. The stacked bar chart shows the distribution (%) of detected transcripts by transcript type. The total number of transcripts is shown above the bars. (**E,F**) Comparison of expression levels of known and novel transcripts. Transcript abundance was measured in reads per million reads (RPM), and log10 converted values for RPM+1 are shown in the violin plot. P-values were calculated by the Wilcoxon rank-sum test. (**E**) Comparison in HCC samples. (**F**) Comparison in non-cancerous liver samples.

reference transcriptome sequences and removed candidates if both results did not detect the same fusion genes (removal of mapping errors). (5) We removed a transcript if the expression level was less than 1% the total amount of the gene expression (removal of possible artificial chimeras). Using these filters, we estimated the expression level of transcripts and identified novel transcript candidates from long-reads, which we classified into transcripts with "novel exon length", "novel exon combination", and "novel exon" (**Fig 1C**).

To evaluate this method, we sequenced cDNA obtained from the MCF-7 cell line and analyzed it with the SPLICE pipeline. As a result, 13,424,802 reads (99.5%) were mapped (**S1 Table**). We calculated the number of reads mapped to each transcript per 1 million reads to estimate the expression levels. The numbers of reads of each transcript were then summed by each gene and compared to the expression level by short-reads sequencing technology [28]. The gene expression levels between long- and short-reads showed a strong correlation ($p < 2.2 \times 10^{-308}$, $r = 0.80$) (**S2 Fig**).

## Comparison of SPLICE method with other tools

We evaluated the detection performance of splicing variants and fusion genes. Although several Nanopore long-reads RNA-seq analysis pipelines exist [21,29–34], to the best of our knowledge, no software can detect both splicing variants and fusion genes. In this study, we used sequence data from MCF-7 and an HCC sample (RK107C) and compared the performance of SPLICE and four other methods (TALON [34], FLAIR [21], StringTie [31], and Bambu [29]) for the splicing variant detection and SPLICE and two other methods (LongGF [32] and JAFFAL [30]) for the fusion gene detection. In the analysis of splicing variants, SPLICE identified the third highest number of transcripts after FLAIR and StringTie (**S3A Fig**). For MCF-7, SPICE had the highest concordance rate with IsoSeq MCF-7 transcriptome data (see Data access) for known transcripts and second highest for novel transcripts (**S3B Fig**).

We next compared the fusion genes detected from MCF7 by SPLICE, LongGF, and JAFFAL. Previously known *BCAS3-BCAS4* [35] was detected only by SPLICE, and *BCAS3-ATXN7* was detected by SPLICE and LongGF (**S4A and S4B Fig**). In HCC, *AC138969.4-PDXDC1* was detected only by SPLICE, *C8B-PAH* was detected by SPLICE and JAFFAL, three fusions (*GCH1-SERPINA6*, *ABCD3-C1orf123*, and *CPS1-WNT10B*) were detected by SPLICE and LongGF, and two fusions (*TBC1D23-TF* and *NBEAL1-RPL12*) were detected only by LongGF. One of the fusions (*NBEAL1-RPL12*), detected only by LongGF, could not be validated by PCR (**S4B Fig**). These results indicate that SPLICE had the highest accuracy for detecting fusion genes. Based on these results, we considered our analysis pipeline to have sufficient accuracy for analyzing transcript aberrations in cancers.

## Analysis of HCC and non-cancerous liver samples

We then analyzed the sequence data from HCCs and matched non-cancerous livers. After filtering low-quality reads (average base quality score < 15), we obtained an average of 9,043,803 reads (12.9 Gbp on average) in the HCCs and 9,013,602 reads (12.4 Gbp on average) in the livers (**S2 Data**). The average mapping rates were 98.9% for the HCCs and 97.3% for the livers (**S2 Data**). Using the SPLICE pipeline, we identified transcripts supported by $\geqq$ 3 reads and obtained a total of 60,011 non-redundant transcripts in the HCCs and 51,506 in the livers (**Fig 1D**). Of these, 66.5% and 70.8% were known protein-coding transcripts, 7.9% and 7.4% were novel protein-coding transcripts, 17.8% and 16.0% were known non-coding transcripts, and 7.8% and 5.8% were novel non-coding transcripts in the HCCs and the livers, respectively (**Fig 1D**). We then compared the expression levels of known and novel transcripts. The average expression levels of novel transcripts were lower than those of known transcripts for both protein-coding and non-coding transcripts in the HCCs and the livers ($p < 2.2 \times 10^{-308}$ for protein-coding, $p < 2.2 \times 10^{-308}$ for non-coding) (**Fig 1E and 1F**).

Among reads mapped to known protein-coding transcripts, 61.7% had full-length coding sequences (CDSs) (**S5A and S5B Fig**). The percentage of reads mapped to the full-length CDS region was negatively correlated with the length of genes (**S5C Fig**). The percentages of reads

containing the full-length CDS were correlated with RIN (RNA Integrity Number) values of the RNA samples ($p = 9.8 \times 10^{-6}$, $r = 0.46$) (**S6 Fig**), indicating the importance of RNA quality for identifying full-length transcripts. We then compared the gene expression levels between previous short-reads and our long-reads data [4] and found high correlations ($p < 2.2 \times 10^{-308}$ in all samples, average $r = 0.79$) (**S7 Fig**).

## Features of novel exons

We identified novel exons by comparing the exon locations in the GENCODE (version 28) and RefSeq (release 88) databases. Our analysis identified 767 and 531 novel exons in 4,769 and 3,786 transcripts of protein-coding genes in the HCCs and matched non-cancerous livers, respectively. Most (92.0%) were the first or last exon of the transcripts (**Fig 2A**). Since TEs are known to be exonized by tissue-specific or cancer-induced hypomethylation of the genome [36], we calculated the percentage of TE-derived novel exons. In the analysis, novel exons > 70% covered by TEs were considered TE-derived novel exons. The percentage of TE-derived novel exons was significantly higher than that of TE-derived known exons in the first, last, and middle exons ($p = 5.9 \times 10^{-153}$ for the first exon, $p = 2.2 \times 10^{-106}$ for the last exon, $p = 1.7 \times 10^{-79}$ for middle exon) (**Fig 2B and S3 Table**). A permutation test showed that the novel first exon was significantly enriched in L2 antisense, ERV1 antisense, and ERVL-MaLR sense (**Fig 2C**). The novel last exon was significantly enriched in *Alu* sense, and the middle exon was significantly enriched in *Alu* antisense, ERVL-MaLR antisense, and ERV1 antisense (**Fig 2C**). These results showed that exonized repeat families and strands differ by their position in the transcripts.

To estimate the biological importance of the novel exons, we estimated their evolutional conservation using a conservation score obtained by 100 species. The average conservation score of the novel exon region was significantly lower than that of the known exon region ($p < 2.2 \times 10^{-308}$) (**Fig 2D**), suggesting that most of the novel exons had less importance in evolutional aspects. However, a part of novel exons was highly conserved (**Fig 2E and S3 Data**), suggesting that they had important biological functionalities.

## Identification of significantly differentiated genes and transcripts

One of the most important features of the long-reads transcriptome is detecting DETs. Thus, by comparing the expression level between HCCs and matched non-cancerous livers, we aimed to find DEGs and DETs. DETs are transcripts that show significant differentiation in expression level between HCCs and matched non-cancerous livers. DEGs are genes obtained by comparing the total amount of transcripts within a gene between HCCs and the livers. We estimated the expression level based on the number of reads and compared them using edgeR software [37]. Our analysis identified 4,744 DEGs and 9,933 DETs after adjusting for multiple testing (FDR < 0.01) (**Fig 3A and S4 Data**) [38]. A Reactome pathway analysis [39] for genes with up-regulated DETs was conducted, and genes with the terms "M Phase", "Cell cycle, Mitotic", and "Cell cycle" were significantly enriched (**S5 Data**). Among the genes with DETs, 746 genes (935 transcripts) were not detected as DEGs (DET-specific genes) (**Fig 3B and S4 Data**). To know the features of the DET-specific genes, we compared the numbers of transcripts identified in this study and transcription start sites (TSSs) of each gene in the database. DET-specific genes had a significantly larger number of transcripts than DEGs ($p = 6.2 \times 10^{-41}$) (**Fig 3C**) and also had a larger number of multiple TSSs (**Fig 3D**) ($p = 5.2 \times 10^{-13}$).

Among the 9,933 DETs, 200 transcripts (99 genes) were from previously reported driver genes (**Fig 3E and S6 Data**) [4,41,42], and 13 genes (16 transcripts) were DET-specific. Of

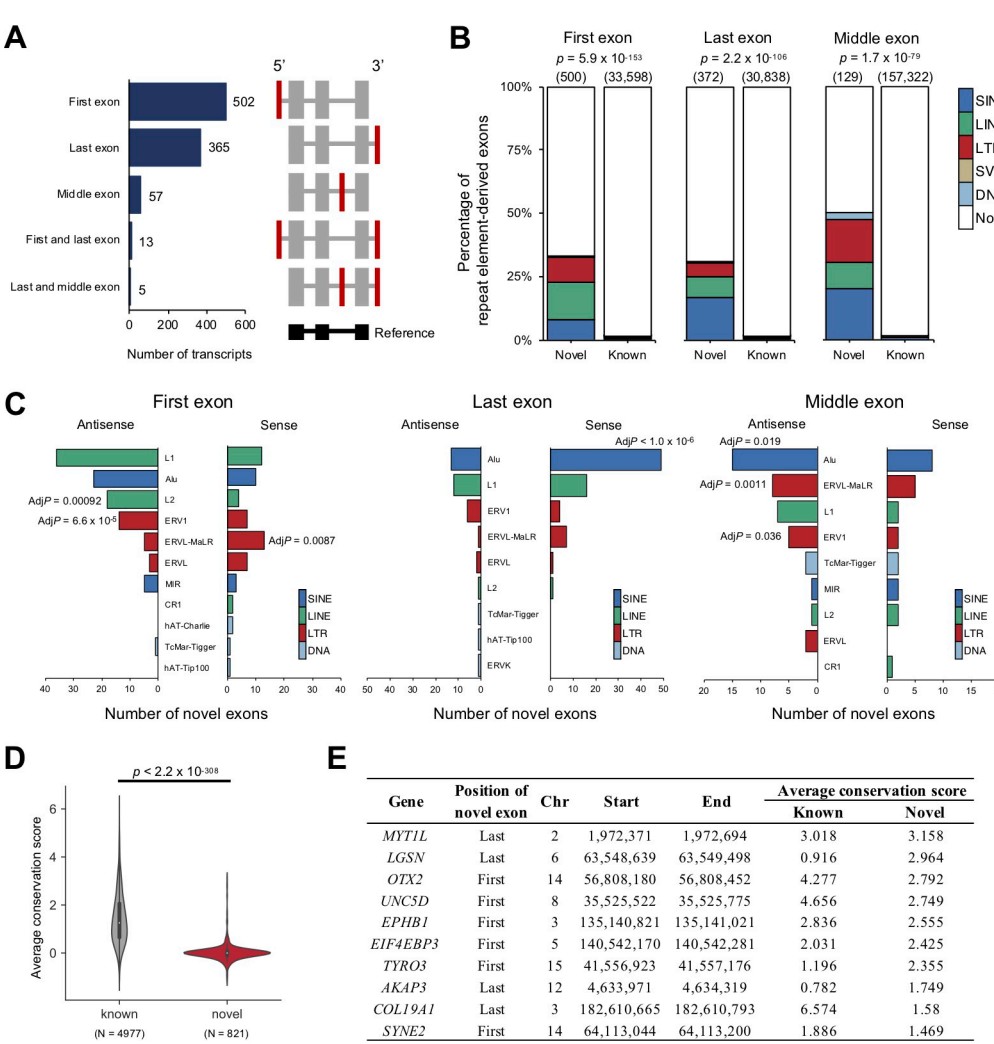

**Fig 2. Features of novel exons.** (**A**) Patterns of transcripts with novel exons. (Left panel) Bar graph showing the number of transcripts by the position of the novel exon. (Right panel) Schematic diagram of the location of novel exons. The novel exons are shown in red. (**B**) Percentage of TE-derived exons in novel and known exons by exon position. The stacked bar chart shows the distribution of TE-derived exons. The total number of exons is shown above the bars. Statistical significance of the total abundance of TE-derived exons between the known and novel exons was calculated by Fisher's exact test. (**C**) The number and enrichment of TE-derived novel exons. TEs were classified by repeat family, and the sense or antisense strand for the gene strand was distinguished. The statistical significance was examined by a permutation test. *P*-values were adjusted by Bonferroni correction (Adj*P*). (**D**) Average conservation scores between novel and known exon regions. For each gene, the average conservation scores were calculated based on the phastCons scores database (100 vertebrates). *P*-values were calculated by the Wilcoxon rank-sum test. (**E**) List of transcripts with novel exons with high average conservation scores. The 10 candidates are shown in order of average conservation score.

these, transcripts of the *MET* oncogene (XM_006715990.2, XM_011516223.1) were up-regulated in HCCs, and tumor suppressor genes, such as *HLA-B* (ENST00000359635.9, ENST00000450871.6) and *APC* (NM_000038.5), were down-regulated (**S6 Data**). Transcripts from 16 HCC driver genes (*ALB, APOB, ASH1L, CDKN2A, CPS1, CTNNB1, DHX9, IDH1, IL6ST, NCOR1, NFE2L2, NRAS, NSMCE2, NUP133, SETDB1,* and *XPO1*) were significantly differentiated (**S6 Data**) [4,41,42]. Of these, transcripts from *ASH1L, CDKN2A, CTNNB1, DHX9, NRAS, NSMCE2, NUP133, SETDB1,* and *XPO1* were up-regulated in the HCCs and other transcripts were down-regulated. *ALB* is known to be highly expressed in the liver, and 3

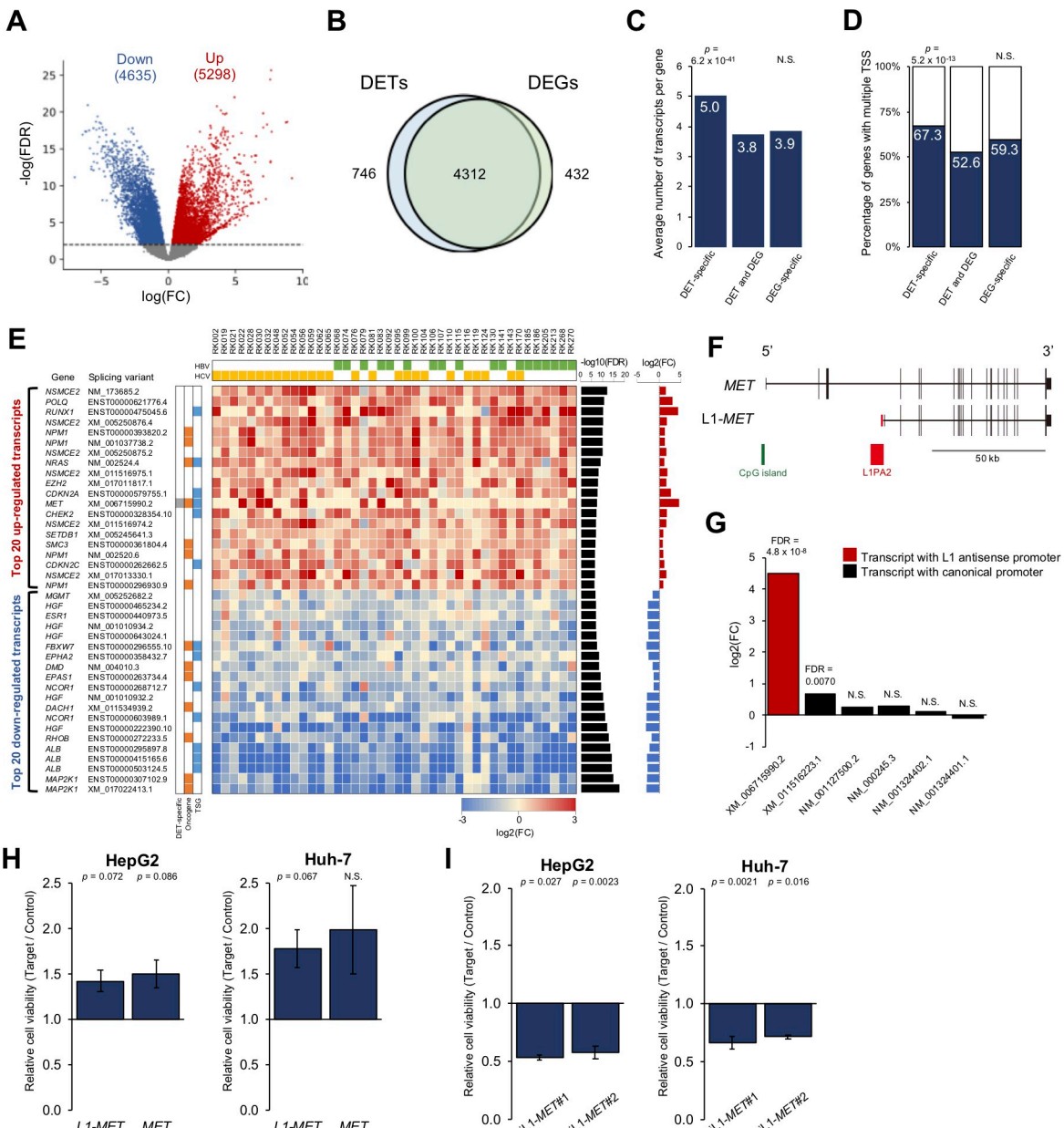

**Fig 3. Landscape of significantly differentiated transcripts between HCCs and matched non-cancerous livers.** (**A**) Volcano plot of DETs in cancers and non-cancerous livers. Red and blue dots represent transcripts that were significantly up- and down-regulated (FDR < 0.01), respectively. (**B**) Venn diagram of the number of genes with DETs and DEGs. (**C**) Comparison of the average number of transcripts per gene for each item in the Venn diagram in **B**. *P*-values for the abundance of transcripts per gene in DET-specific and DEG-specific genes were calculated by the Wilcoxon rank-sum test. The numbers of transcripts in each gene were compared for DET-specific genes and other genes and for DEG-specific genes and other genes. (**D**) Percentage of genes with multiple TSS for each item in the Venn diagram in **B**. *P*-values for the enrichment of genes with multiple TSS in DET-specific and DEG-specific genes were calculated by Fisher's exact test. The numbers of TSS in each gene were compared for DET-specific genes and other genes and for DEG-specific genes and other genes. (**E**) The landscape of DETs in driver genes. Twenty up-regulated and down-regulated DETs are shown in order of lower FDR. Heatmap showing the expression differences (log2(fold change)) in cancer and matched non-cancerous livers. (**F**) Schematics diagram of the structural differences in L1-*MET* (XM_006715990.2) and other *MET* transcripts. L1-*MET* has the L1-derived first exon shown in red. (**G**) Bar chart showing expression differences between cancers and matched livers in *MET* transcripts. The red bar represents the *MET* transcript with the L1 antisense promoter, and black bars represent *MET* transcripts with the canonical promoter. *P*-values were obtained using quasi-likelihood methods in EdgeR software [40]. The Benjamini-Hochberg method was used for multiple testing correction (FDR). (**H**) Effect of L1-*MET* overexpression on the proliferation of HepG2 and Huh-7 cell lines assessed by CCK-8. *P*-values were calculated by the one-sample t-test (n = 3). (**I**) Effect of L1-*MET* knockdown on the proliferation of HepG2 and Huh-7 cell lines assessed by CCK-8. *P*-values were calculated by the one-sample t-test (n = 3). Error bars represent the standard error of the mean. N.S.: Not significant.

major transcripts (ENST00000503124.5, ENST00000415165.6, and ENST00000295897.8) were detected in HCCs and matched livers (**S6 Data**).

We then focused on genes that had both significantly up-regulated and down-regulated transcripts (referred to as bi-directionally significantly expressed (BiExp) genes). Our analysis identified 80 BiExp genes (270 transcripts) (**S7 Data**). Among them, *AFMID*, which is a previously reported oncogenic BiExp gene [15], was successfully detected in our analysis (**S8 Fig**). Among the 746 DET-specific genes, 42 were BiExp genes. As expected, DEGs had a smaller percentage of BiExp genes than did DET-specific genes ($p = 1.6 \times 10^{-15}$) (**S4 Table**), as BiExp genes had both up- and down-regulated transcripts whose expression levels canceled out each other.

## Exonization of TEs in HCCs

In the HCCs, a transcript of *MET* (XM_6715990.2) was ranked first among the significantly up-regulated DET-specific driver genes (FDR = $4.8 \times 10^{-8}$) (**Fig 3E** and **S6 Data**). A previous study reported that the first exon of XM_006715990.2 is L1-derived, and this transcript is expressed by the L1 antisense promoter (referred to as L1-*MET*) (**Fig 3F**) [43]. L1-*MET* was remarkably up-regulated in HCCs compared to other *MET* transcripts with the canonical promoter (**Fig 3G**). Since the role of L1-*MET* in HCCs is largely unknown, we performed overexpression and knockdown experiments of L1-*MET* in the HepG2 and Huh-7 cell lines. The overexpression of L1-*MET* and canonical *MET* (NM_001127500.2) increased cell proliferation (**Figs 3H** and **S9**), and the knockdown decreased cell proliferation of both cell lines (**Figs 3I** and **S10**), indicating that L1-*MET* contributes to the proliferation of HCC cells. Since the exonization of TEs was observed in HCCs and livers (**Fig 2B**), we next focused on cancer-specific TE-derived exons (n = 829) (**S5 Table**). We found that they were significantly enriched in the first and last exons ($p = 7.6 \times 10^{-43}$ for the first exon, $p = 3.9 \times 10^{-7}$ for the last exon) (**S11 Fig**). An analysis of repeat families showed that L1 sense and antisense, L2 antisense, ERVL sense and antisense, ERVL-MaLR sense, and LTR sense were significantly enriched in the first exon of cancer-specific transcripts (**S5 Table**). The liver-specific TE-derived exons were also significantly enriched in the first and last exons ($p = 2.9 \times 10^{-6}$ for the first exon, $p = 0.0012$ for the last exon) (**S11 Fig**), and by repeat families, the L2 antisense was significantly enriched in the first exon ($p = 8.6 \times 10^{-5}$) (**S6 Table**).

## Expression from HBV

We then analyzed reads containing the HBV genome sequence in 18 HBV-positive patients. Of these samples, 13 HCCs and 10 matched non-cancerous livers expressed transcripts having the HBV genome sequence (HBV transcript) (**S12 Fig**). We classified transcripts having the HBV genome sequence into HBV-only and HBV-human fusion transcripts. We estimated transcriptional start and end sites and analyzed these locations on the HBV genome. Most of the TSSs of HBV transcripts were clustered in the upstream region of the PreS2 gene (around 1 bp– 100 bp in the HBV reference genome) (**Figs 4A** and **S12**) as shown in a previous study using the CAGE method [44]. We then compared the expression levels of the transcripts with the HBV sequences. Expression levels were estimated by log10 converted support reads per million reads (log10(RPM)) for each sample. The expression levels of each HBV transcript were significantly different between HCCs and the livers ($p = 0.046$) (**Fig 4B**); however, the comparison of the total amounts of HBV transcript was not (**S13A Fig**), suggesting that the transcript-level analysis with long-reads can detect significantly differentiated transcripts from HBV.

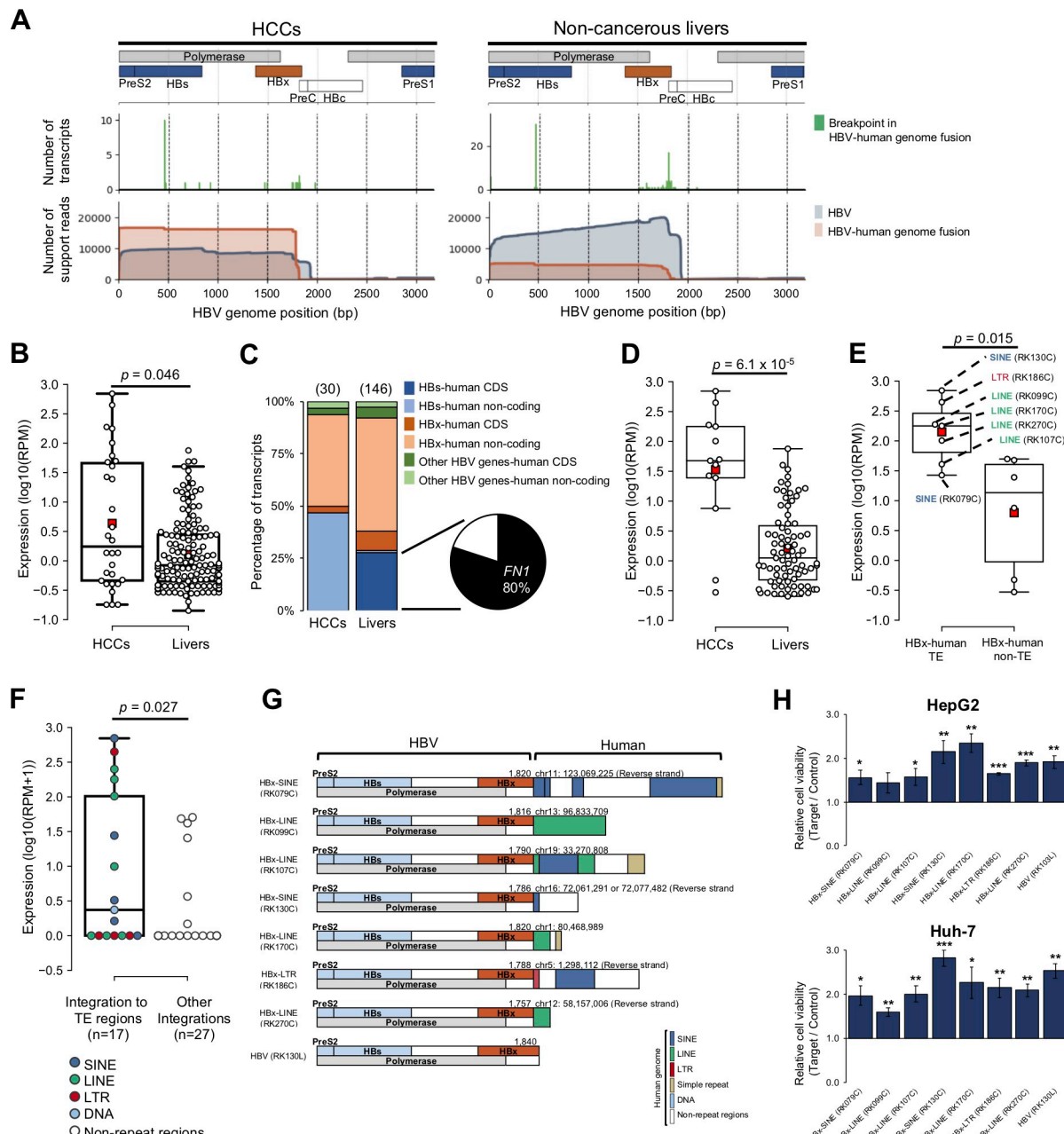

**Fig 4. Identification of HBV-human genome fusion transcripts.** (**A**) Visualization of the RNA-seq coverage in the HBV genome region in HCCs and non-cancerous livers. (Upper panels) Schematic diagram of the HBV genome structure. (Middle panels) Bar chart of the breakpoints in HBV-human fusion transcripts. (Lower panels) Coverage of HBV transcripts (HBV alone) and HBV-human fusion transcripts are shown in blue and red. The y-axis shows the total number of support reads (raw counts). (**B**) Expression levels of HBV-human fusion transcripts in HCCs and livers. Transcript abundance was measured in reads per million reads (RPM), and log10 converted values for RPM+1 are shown in the boxplot. *P*-values were calculated by the Wilcoxon rank-sum test. (**C**) Percentage of HBV-human fusion transcripts in HCCs and livers. The stacked bar chart shows the distribution (%) of detected transcripts by type. The total number of transcripts is shown above the bars. The pie chart shows the percentage of HBs-*FN1* fusion transcripts in HBs-human CDS in the livers. (**D**) Expression levels of HBx-human non-coding fusion transcripts in HCCs and livers. *P*-values were calculated by the Wilcoxon rank-sum test. (**E**) Expression levels of HBx-human TE fusion and HBx human non-TE fusion transcripts in HCCs. *P*-values were calculated by the Wilcoxon rank-sum test. (**F**) Expression levels of HBV transcripts integrated into TE regions and those integrated into non-TE regions. *P*-values were calculated by the Wilcoxon rank-sum test. (**G**) Schematic diagram of HBx-human TEs fusion transcripts. (**H**) Effect of HBx-TE fusion on the cell proliferation of HepG2 and Huh-7 cell lines assessed by CCK-8. Error bars represent the standard error of the mean. *P*-values were calculated by the one-sample t-test ($^*p < 0.1$, $^{**}p < 0.05$, $^{***}p < 0.01$) (n = 3).

Thirty and 146 HBV-human genome fusion transcripts were detected in the HCCs and matched livers (**S8 Data**), and all were selectively expressed in the respective tissues. Breakpoints of HBV-human fusion transcripts were clustered in the HBs gene region (450 bp– 470 bp) and downstream of the HBx gene region (1750 bp– 1840 bp) (middle panel of **Fig 4A**). Therefore, we focused on HBs and HBx genes and classified HBV-human fusion transcripts into "HBs-human CDS fusion", "HBs-human non-coding fusion", "HBx-human CDS fusion", "HBx-human non-coding fusion", "Other HBV genes-human CDS fusion", and "Other HBV genes-human non-coding fusion" (**Fig 4C**). Forty HBs-human CDS fusion transcripts were detected in the livers but not in the HCCs (**Fig 4C**). Among them, 32 (80.0%) were HBs-*FN1* fusion transcripts (**Fig 4C**) [45]. Expression levels were then compared between the HCCs and matched livers in these categories (**Figs 4D and S13A**). HBx-human non-coding fusion showed higher expression levels in the HCCs than in the livers ($p = 6.1 \times 10^{-5}$) (**Fig 4D**); however, other categories did not show significant differences (**S13B Fig**). Previous studies reported that LINE1(L1), a retrotransposon, forms the HBx-L1 fusion transcript and induces carcinogenesis in HCCs [46,47]. We thus focused next on HBx-human TE fusion and identified 4 HBx-LINE fusion transcripts, 2 HBx-SINE fusion transcripts, and 1 HBx-LTR fusion transcript in HCCs (**S13C Fig**). The expression level of HBx-human TE fusion was significantly higher than that of HBx-human non-TE fusion in HCCs ($p = 0.015$) (**Fig 4E**).

The expression pattern of HBx-human TE fusions suggests their important roles in carcinogenesis. Therefore, we examined a possible mechanism for the promotion of expression and their functional effect on cell proliferation. Our samples in this study have been used in a previous WGS study, and genomic locations of HBV integrations have been determined [4]. The WGS identified 44 HBV integration sites, and 17 of them were integrated into TE regions. A comparison of their expression levels showed that integrations into the TE regions result in a higher expression than into the non-TE regions ($p = 0.027$) (**Fig 4F**). We then analyzed the role of HBx-human TE fusions on cell proliferation with an overexpression experiment of full-length HBx-TE fusions and canonical HBx in liver cancer cell lines (HepG2 and Huh-7) (**Figs 4G and S9**). We found that HBx-TE fusions significantly increased cell growth in both cells (**Fig 4H**).

## Fusion genes

We detected 164 cancer-specific fusion transcripts (**S9 Data**). Although recurrent fusion events of the same gene pairs were not detected, six genes, *SENP6*, *TBRG1*, *SDC2*, *ABCD3*, *LGSN*, and *LDLR*, appeared recurrently but fused with different partners (**Fig 5A**). We experimentally validated all candidates (12/12) by RT-PCR (**Fig 5B and S7 Table**). Within the samples of the current study, 15 have been subjected to fusion gene detection by short-reads in the ICGC pan-cancer study [24] (**S14A Fig**). A comparison between the short- and long-reads revealed that 41 out of 50 fusions (82.0%) were not detected in the short-reads and only 9 fusions (8.0%) were commonly detected (**Fig 5C**). We considered three explanations for the low overlap. (1) Since most of the fusion genes missed by short-reads had very low expression levels (**S14B Fig**), many fusion genes with low expression levels were missed by the single approach. (2) In addition, 28.9% of transcripts in long-reads lacked the 5' region (**S5 and S14C and S14D Figs**). Therefore, fusion genes whose breakpoints are located in the 5' region are difficult to detect using long-reads. (3) Since artificially truncated cDNA can be produced by internal poly-A priming without priming to poly-A at the 3'end of the mRNA, fusion genes with breakpoints in the 3'region may also be missed. This comparison suggests that a combination of short- and long-reads can detect more fusion genes.

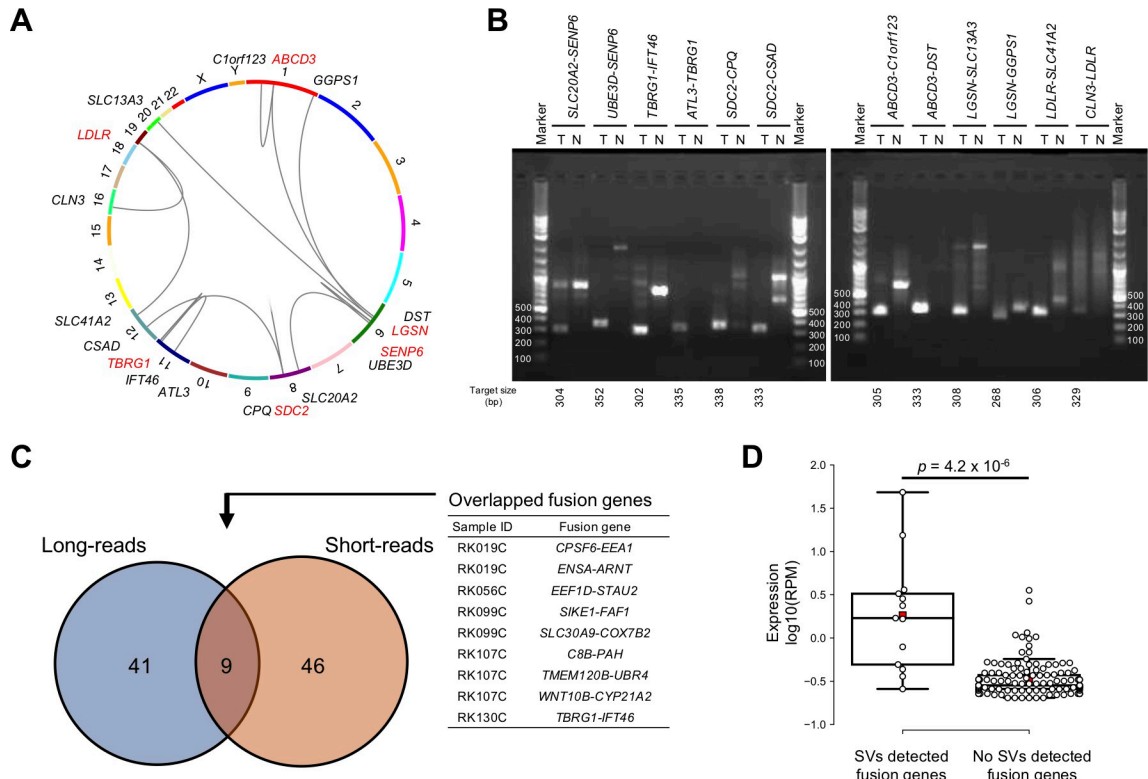

**Fig 5. Identification of fusion genes.** (**A**) Circos plot of recurrent fusion genes. The genes shown in red are recurrent fusion genes. (**B**) PCR validation of the recurrent fusion genes. The results of an electrophoresis of each fusion gene in cancer (T) and matched non-cancerous livers (N) are shown. (**C**) Venn diagram of fusion genes detected by long- and short-reads sequencers. We compared 15 HCC samples in which short-reads RNA-seq was analyzed in the ICGC study [24]. (**D**) Expression levels between fusion genes supported and not supported by SVs according to WGS. *P*-values were calculated by the Wilcoxon rank-sum test.

We also compared the fusion genes and structural variations (SVs) detected by previous WGS [4,23]. Among the 164 fusion transcripts, 21 had SVs that can cause the fusion transcripts, and their expression levels were significantly higher than other fusions ($p = 4.2 \times 10^{-6}$) (**Fig 5D**). This result suggests that there are many sub-clonal fusion genes, and their expression levels are lower than clonal fusion genes. Although the functional importance of sub-clonal fusion genes is currently unknown, deeper RNA-seq would detect more of them.

## Expression profiles of HBV- and HCV-infected HCCs

Our non-cancerous liver samples have hepatitis by infection with HBV or HCV. To examine the influence of HBV and HCV on hepatitis and HCC, we performed a clustering analysis for 16 HBV-infected, 24 HCV-infected, and 2 dually infected samples in HCCs and non-cancerous livers, respectively (**Fig 6A**). In the non-cancerous livers, most HBV- and HCV-infected tissues were in separate clusters, whereas the HCCs were not separated (**Fig 6A**). Next, we analyzed genes with significantly differentiated transcripts between HCV and HBV samples. In the HCV- and HBV-infected livers, 284 and 68 transcripts were significantly up-regulated, respectively (**Fig 6B and 6C**). Highly expressed transcripts in the HCV-infected livers were significantly enriched in immune system-related pathways, especially in the interferon signaling pathway (**Fig 6D** and **S8 Table**). Transcripts up-regulated in the HBV-infected livers had no significantly enriched pathways. For the HCV- and HBV-infected HCCs, we performed the same analysis and found that immune system-related pathways were slightly enriched in DETs

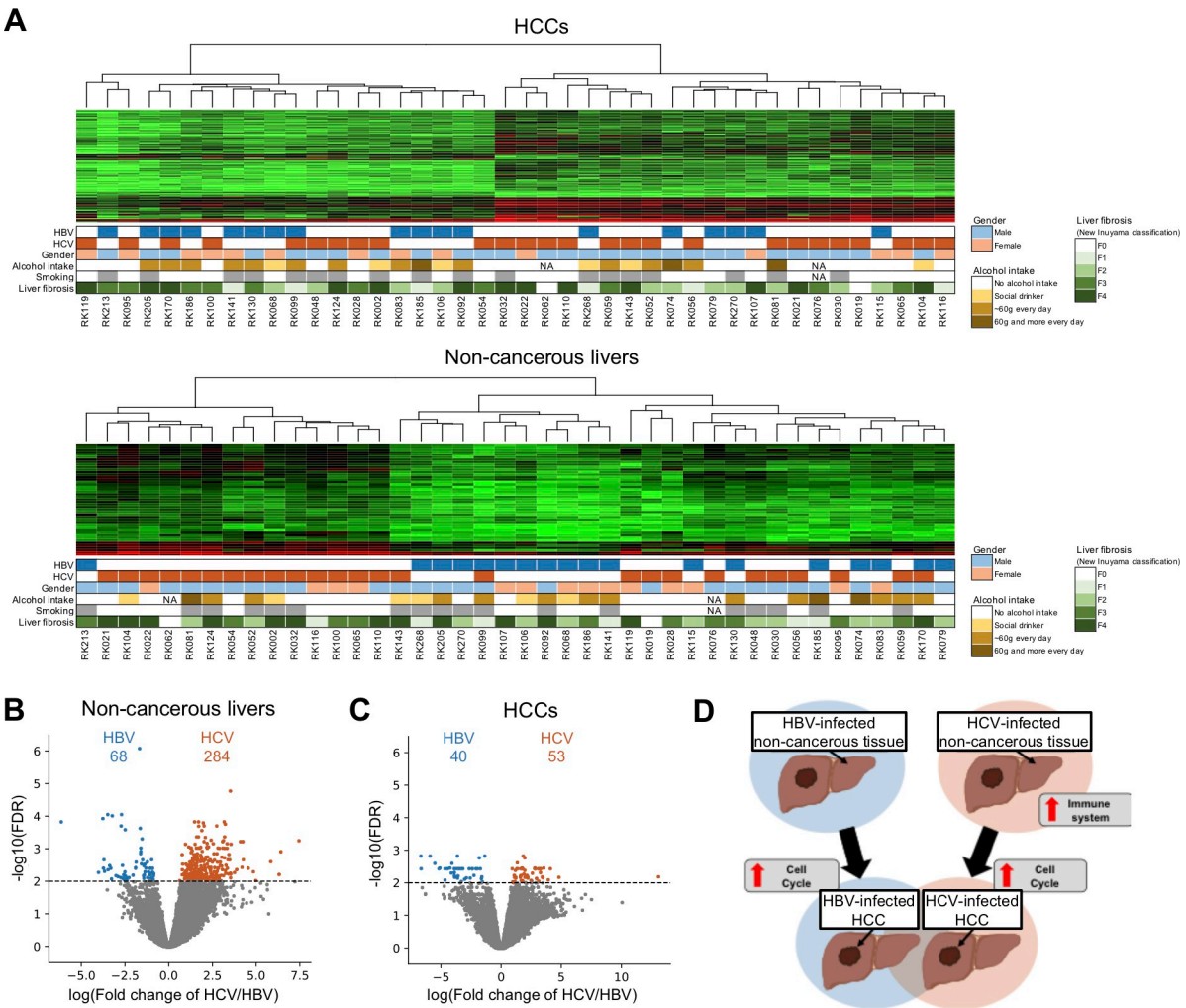

**Fig 6. Comparison of expression profiles of HBV- and HCV-infected HCCs.** (**A**) Clustering by the transcriptome of 16 HBV-infected tissues, 24 HBV-infected tissues, and 2 dually infected tissues in HCCs (upper) and in matched non-cancerous livers (lower). Clinical and pathological information of the samples is shown in the lower panels. (**B, C**) Volcano plots of DETs in HBV- and HCV-infected tissues. Light blue and orange dots represent transcripts that were significantly up-regulated in HBV- and HCV-infected tissues with FDR < 0.01. (**B**) Comparison in non-cancerous liver samples. (**C**) Comparison in HCC samples. (**D**) Schematic diagram of the expression profiles of HBV- and HCV-infected tissues.

only in the HCV-infected HCCs, but not as pronounced as in the non-cancerous livers (**Fig 6C and S8 and S9 Tables**). Genes of cell cycle-related pathways were significantly up-regulated in both HBV- and HCV-infected HCCs, as in all HCCs (**Fig 6D and S5 and S10 and S11 Data**).

## Discussion

In this study, we used a long-reads sequencer (Oxford Nanopore) for a comprehensive analysis of transcriptional abnormalities in HCCs and non-cancerous livers. We first developed an analysis pipeline (SPLICE) to analyze full-length transcripts (**Fig 1A and 1B**). SPLICE removed possible errors around splicing junction sites and pseudo-fusion transcripts by considering alignment and mapping errors (**Fig 1B**). To our knowledge, SPLICE is the only long-reads RNA-seq pipeline that can detect both splicing variants and fusion genes. While most long-reads RNA-seq analysis pipelines for detecting splicing variants use short-reads RNA-seq

data for the alignment correction of splice junctions [21,31,34], SPLICE considers error rates around splicing sites (**S3C Fig**) and can obtain accurate results using only long-reads RNA-seq data (**S3 Fig**). SPLICE also identified a larger number of fusion genes than existing software (**S4 Fig**). A comparison with short-reads showed high concordances in the estimation of gene expression levels, suggesting that our analysis was reliable (**S7 Fig**).

In the analysis of HCCs and matched livers, 15.7% and 13.2%, respectively, of transcripts were not found in the database and considered novel (**Fig 1D**). The expression levels of the novel transcripts were significantly lower than those of known transcripts, and deeper sequencing is expected to detect a larger number of novel transcripts (**Fig 1E and 1F**). Our analysis identified novel exons. the majority of which were first and last exons (**Fig 2A**). The novel first exons can be produced by TSSs from unknown alternative promoters and may contribute to tissue-specific or disease-associated mRNA regulation. Novel last exons can cause new UTRs and may affect the stability of mRNA. Since the average conservation scores of novel exons were lower than those of known exons, most novel exons are assumed to not have important functions (**Fig 2D**). However, some novel exons were highly conserved and could have essential functions for biological activities (**Fig 2D and 2E**). In the *MYT1L* gene, for example, a highly-conserved novel exon was detected (**Fig 2E**), and only a transcript with the novel exon was expressed. This transcript was significantly down-regulated in HCCs (**S15 and S16 Figs**). Previous studies reported that the expression of *MYT1L* is neuron-specific [48] and has been suggested to play a critical role in neuronal differentiation and maintenance [48,49]. Our analysis identified a new transcript with the novel exon of this gene in the liver. Several studies suggested that *MYT1L* suppresses the proliferation of glioblastoma, and this transcript could have tumor-suppressive effects in HCCs as well [50,51].

This study also identified the exonization of TEs. Interestingly, 30–50% of novel exons in protein-coding genes were derived from TEs (**Fig 2B**), suggesting that TEs can be a source of the functional diversity of genes [52]. Strands of the exonized TEs were not random, and patterns were different among the location of the exons (**Fig 2C**); first and middle exons had higher percentages of TEs encoded on opposite (antisense) strands, whereas the opposite pattern was observed in TEs of the last exons (**Fig 2C**). A previous study suggested that the antisense of LINE1 has a promoter function and can start transcription [43,53]. This observation can partly explain the strand bias of TEs of first exons. The strand bias of TEs of the middle exon can be caused by the asymmetric distribution of TEs in introns. TEs on sense strands generate harmful poly-A signals in introns, resulting in the reduction of TEs in sense strands in introns [54,55]. *Alu* has polyadenylation signals in the sense strand that can generate the last exons, which can explain the strand bias of *Alu* of the last exon (**Fig 2C**) [56].

A comparison of expression levels between HCCs and non-cancerous livers showed the differentiation of several transcripts with TEs. A transcript with a TE-derived exon from an important oncogene, *MET*, (L1-*MET*) was significantly overexpressed in HCCs (**Fig 3F and 3G**). L1-*MET* was reported to be involved in the progression of colorectal cancer and bladder cancer [43,57], and this transcript also promoted the proliferation of HCCs (**Fig 3H and 3I**). Since L1-*MET* is suggested to be regulated by the L1 antisense promoter (**Fig 3F**) [43,53,57], its expression is independent of other *MET* transcripts. Indeed, 6 transcripts were observed in *MET*; however, only L1-*MET* was highly overexpressed in HCCs (**Fig 3G**). Another oncogenic TE fusion event was found in the *HRH1* gene (L2-*HRH1*). Our study identified 3 transcripts from the *HRH1* gene; however, only L2-*HRH1* (NM_001098211.1) was significantly upregulated in HCCs, and the overexpression of L2-*HRH1* promoted cell proliferation (**S17 Fig**). The *HRH1* gene is known to promote the progression of HCCs [58], and our finding of the fusion events suggests an activation mechanism of the *HRH1* gene. Additionally, TE fusion transcripts were identified in HBV-infected HCCs (HBx-TE fusion) (**Fig 4G**). The overexpression

of HBx-TE fusion increased cell proliferation in the cell lines (**Fig 4H**), suggesting that HBx-TE has a similar function as canonical HBx [59]. Since the expression levels of the integrated HBV to the TE regions were significantly higher than those of the non-TE regions (**Fig 4F**), integrations into TE regions may cause the constitutive expression of HBV genes via TE promoters. These results strongly suggest the important role of TEs on gene expression. Most TEs are repressed by several inactivation mechanisms; however, epigenetic changes in cancer cells can activate TEs as promoters, resulting in the aberrant regulation of oncogenes [60].

Most RNA-seq studies identified DEGs but far fewer DETs. In this study, we detected 9,933 DETs (**Fig 3A**). Seven hundred forty-six genes with DETs were not detected by the gene-level analysis (DET-specific genes), indicating that the analysis of transcripts benefits the detection of novel driver genes (**Fig 3B**). Our analysis also showed possible mechanisms for regulating splicing variants. DET-specific genes had a higher number of transcripts per gene and a higher percentage of transcripts with multiple TSSs (**Fig 3C and 3D**). This observation suggests that multiple transcripts regulated by different promoters canceled each other out, and thus expression differences were not detected in the gene-level analysis. We also found 80 genes with both significantly up-regulated and down-regulated transcripts (BiExp genes) (**S7 Data**). In addition to the previously reported *AFMID* gene [15], genes related to aberrant cell proliferation (*CEACAM1*, *CASP8*, *LDHB*, *SPINT2*) [61–64] were detected as BiExp genes (**S7 Data**). Each transcript in BiExp genes is regulated differently and have different functions.

We then compared the expression level of transcripts in HCCs and non-cancerous livers upon HBV or HCV infection (**Fig 6**). A clustering analysis did not separate HBV- and HCV-infected HCCs, whereas the livers were clustered into three groups: HCV cluster, HBV cluster, and cluster with both types (**Fig 6A**). This observation suggests that the difference in virology mainly affects the phenotype of the livers (**Fig 6D**). A pathway analysis for genes differentiated between HCV- and HBV-infected livers showed an enrichment of immune-related genes (**S8 Table**). This result is consistent with previous studies [65,66] and suggests that HBV and HCV may induce different expression changes but that similar mechanisms work after cancer development (**Fig 6A** and **S8** and **S9 Table** and **S10** and **S11 Data**).

Although we obtained many interesting discoveries by analyzing full-length transcripts, our study has limitations that should be addressed in future studies. First, our analysis did not determine the structure of the entire CDS for 38.3% of the mapped reads. The majority of the incomplete transcripts lacked 5' regions (**S5B and S5C Fig**), which can be explained by the process of cDNA synthesis and degradation of RNA samples. cDNA synthesis starts from the 3'end and, therefore, the inactivation of reverse transcriptase or degradation of RNA samples results in cDNA lacking the 5' end (**S6 Fig**). This mechanism can also affect fusion-gene detection (**Figs 5 and** S14C and S14D). Improvements in cDNA synthesis and the use of high-quality RNA can increase the rate of full-length transcripts. Second, the accuracy of the analysis depends on the sequencing error rate. Although several filters were applied to currently available high-error reads (**Figs 1B and S1**), sequencing errors can still affect the accuracy of the result. With improvments in sequencing technologies and basecallers, the identification of larger numbers of splicing changes with high accuracy should be possible in the future (**S10 Table**). Third, our study identified a higher expression of HBx-human TE transcripts that promoted cell proliferation (**Fig 4G and 4H**). However, the detailed biological mechanisms of the transcripts are not clear. A previous study reported that HBx-LINE1 can change the expression levels of epithelial-mesenchymal transition markers [47]; however, our analysis did not detect significant changes in these genes (**S18 Fig**). Since the HBx-human transcripts may, at least in part, explain the carcinogenesis of liver cancer, further functional analysis is needed.

To conclude, here we developed the SPLICE pipeline for transcriptome analysis using a long-reads sequencer. Validation with the MCF-7 cell line and PCR showed the reliability of

our analysis. Using this method, we analyzed the full-length transcript of HCCs. We detected 746 genes with DETs, which were otherwise difficult to detect by short-reads sequencers. We also detected chimeras of TEs and cancer-related genes and found that they promote the proliferation of HCC cell lines. In the analysis of fusion genes, we found 6 recurrent genes that fused with other genes. Additionally, by comparing HBV- with HCV-infected livers, a significant difference in immune-related gene expression levels was detected. Our results strongly suggest that the direct observation of transcripts with long-reads sequencing contributes to understanding the true picture of transcript aberration in cancer.

## Methods

### Ethics statement

All subjects gave informed consent to participate in the study following International Cancer Genome Consortium (ICGC) guidelines. Institutional review boards at Institute of Physical and Chemical Research (RIKEN) and the University of Tokyo, and all groups participating in this study approved this work (RIKEN; H20-11(16), the University of Tokyo; 2020097G). All participants have provided written consent to participate in the study.

### Samples

This study was conducted following the principles of the Declaration of Helsinki. RNA from HCCs and adjacent non-cancerous livers from 42 patients was used for this study. These samples have been used for short-reads sequencing and reported by several previous studies (**S1 Data**) [4,23,24], and a list of somatic mutations (point mutations, short indels and structural variations), gene expression levels, and fusion genes by short-reads analysis are available [4,24]. RNA sample qualities were evaluated with RIN (RNA Integrity Number) values using Bioanalyzer (Agilent), and samples with RIN values $\geq$ 8 were selected. RNA from 16 HBV-infected HCCs, 24 HCV-infected HCCs, 2 dually infected HCCs, and their matched non-cancerous livers were sequenced.

### Library preparation and cDNA sequencing

Full-length cDNA was synthesized from total RNA (1 μg) using the SMARTer PCR cDNA Synthesis kit (Clontech) according to the manufacturer's instructions. For primer digestion, cDNA samples were treated with exonucleaseI(NEB) at 37˚C for 30 min. After purification with Agencourt AMPure XP magnetic beads (Beckman Coulter), libraries were constructed using the Ligation Sequencing Kit (SQK-LSK109) (Oxford Nanopore) according to the manufacturer's protocol. Libraries were sequenced on a SpotON FlowCell MKI(R9.4) (Oxford Nanopore) using the MinION sequencer (Oxford Nanopore). Basecalling from raw data (FAST5 format) was performed using Guppy software (version 3.0.3) (Oxford Nanopore).

### Analysis pipeline

We constructed an analysis pipeline named SPLICE (**Fig 1A**). Low-quality reads (average base quality < 15) were filtered out. Reads were mapped to the reference genome sequence (hg38), the reference transcriptome sequence (GENCODE (version 28), and RefSeq (release 88)) using minimap2 software (v2.17) [27]. For reads mapped to the reference genome sequence, a $\geqq$ 60-bp unmapped region (soft-clipping regions) was re-mapped to the reference genome sequence (hg38). Since sequencing errors near splicing junctions can lead to errors in predictions of the splicing sites, we compared the mapping results to the reference genome and the

reference transcriptome sequence and removed reads if the number of matches to the reference genome sequence was smaller than that to the reference transcriptome sequence.

In the present study, we analyzed sequence data from MinION, but our method is applicable to data from PromethION and flongle as well.

### Identification of known transcripts

The genomic locations of known transcripts were obtained from GENCODE (version 28) and RefSeq (release 88), and transcripts that were commonly registered in both annotations were unified to GENCODE transcript IDs (**Fig 1A**). We assigned each exon in the transcripts to the exons in the database if the positions of the splicing junction sites at both ends of the transcript's exon and reference exon were perfectly matched. We then considered the combination of the exons and transcript ID in the database were assigned. Transcripts that were not found in the database were classified as novel. For the annotation, we did not consider UTR lengths for the classification, because the qualities of the ends of the reads were low. Reads that contained all coding exons of transcripts in GENCODE or RefSeq were classified as "full-length CDS". When reads had a part of the exons of a known transcript but did not cover all the exons, they were classified as "partial-length CDS". Among partial-length CDSs, if the reference transcripts had two or more exons and reads had only one exon, we removed the candidates as highly truncated reads. For partial-length CDSs reads annotated to multiple known transcripts, we annotated them to the longest known transcript among matched known transcripts. After the annotation, we removed transcripts if the percentage of the expression level of a transcript was less than 1% the total expression of the gene.

### Identification of novel transcripts

Reads that were not annotated as known transcripts were classified as "novel exon length", "novel exon combination", or "novel exon" (**Fig 1C**). "Novel exon length" reads contain known exons with different lengths from the reference database. As described above, a change of splicing junction sites $\geq$ 5-bp was considered a length change. If splicing junction sites changed $<$ 20 bp for a known splicing junction site, we evaluated the mismatch rate within $\pm$ 5 bp regions from splicing junction sites and removed the candidates for error rates $\geqq$ 20%. Novel exons were defined as expressed regions that were not overlapped with known exons. Novel exon candidates were removed if they were expressed only in a single novel exon. We removed candidates if the percentage of their transcript expression level was $<$ 1% the total expression level of the gene.

### Identification of fusion genes

Reads mapped to two protein-coding genes were considered fusion gene candidates. Possible errors in fusion gene candidates were filtered out as follow. (1) Because mapping errors due to highly homologous genomic regions can cause fusion gene candidates, we compared the results of mapping to the reference genome and the reference transcriptome sequences and removed candidates if both were inconsistent (**Fig 1B**). (2) Because fusion gene candidates can be generated by mis-ligation of two transcripts during the library preparation, we removed candidates that contained primer sequences for cDNA synthesis between two genes (**Fig 1B**). (3) Because artificial chimeric reads may be generated from highly expressed genes, there is a higher chance of mis-ligation in the library preparation; thus, we removed candidates if their percentage of transcript expression level was $<$ 1% the average expression level of both genes. (4) Candidates with low expression levels (support reads $<$ 3) in two or more samples in the matched non-cancerous livers were removed as artifacts. (5) Candidates were removed as

read-through transcripts if the two transcripts were the same strand and were in close proximity to each other in the genome (< 200,000 bp).

We further compared fusion genes with SVs detected by previous WGS [4,23]. If the breakpoints of an SV were located in intron or exon regions of the fusion gene, the SV was considered a possible cause of the fusion gene.

## Benchmarking

To evaluate the performance of SPLICE, we compared it to four widely adopted long-reads RNA-seq analysis pipelines: TALON [34], FLAIR [21], StringTie [31], and Bambu [29]. For this comparison, we used MCF-7 and HCC (RK107C) sequence data. We randomly extracted 1 M reads from the MCF-7 and HCC (RK107C) sequence data using Seqtk (v1.3) (params: sample -s1 1000000). Reads were mapped to the reference genome sequence (hg38) with minimap2 (v2.17) (params: -ax splice—MD), and the output SAM files were converted to BAM files and sorted with samtools (v1.7) [67]. GENCODE v28 and RefSeq release 88 annotation databases were used in all data processes.

For the benchmarking of TALON (v5.0), we corrected aligned reads with TranscriptClean (v2.0.3) [68]. Next, we ran the *talon_label_reads* module to flagging reads for internal priming (params:—ar 20). The TALON database was initialized by running the *talon_initialize_database* module (params:—l o —5p 500 —3p 300). Then we ran the *talon* module to annotate the reads (params:—cov 0.8—identity 0.8). To output transcript abundance, we first obtained a whitelist using the *talon_filter_transcripts* module (params:—maxFracA 0.5—minCount 5) and then performed transcript quantification using the *talon_abundance* module based on the whitelist.

For the benchmarking of FLAIR (v1.5), the sorted BAM file was converted to BED12 using *bin/bam2Bed12.py*. Then we corrected misaligned splice sites with the *flair-correct* module. High-confidence isoforms were defined from the corrected reads using the *flair-collapse* module (params: -s 3—generate_map).

For the benchmarking of StringTie (v2.2.1), Stringtie was performed with input files consisting of long-reads alignments and reference annotations (params: -L -c 3).

For the benchmarking of Bambu (v2.0.0), Bambu was performed with input files consisting of long-reads alignments, reference annotations, and the reference genome (hg38) (params: min.readCount = 3).

Candidates with low expression levels (support reads < 3), antisense, and intergenic transcripts were excluded from the comparison.

## Assessing the performance of fusion gene detection in SPLICE and other tools

To compare the fusion genes detected by SPLICE, LongGF [32] and JAFFAL [30], we used 1 M reads randomly extracted from the MCF-7 and HCC (RK107C) sequence data as described above.

For the benchmarking of LongGF (v0.1.2), reads were mapped to the reference genome sequence (hg38) with minimap2 (v2.17) (params: -ax splice—MD), and the output SAM files were converted to BAM files and sorted with samtools (v1.7). We then ran the *longgf* module and obtained the list of fusion genes (params: min-overlap-len 100 bin_size 50 min-map-len 200 pseudogene 0 secondary_alignment 0 min_sup_read 3).

For the benchmarking of JAFFAL (v2.2), we ran the *JAFFAL.groovy* module with zipped fastq files.

In this comparison, close gene pairs ($< 200,000$ bp) were not considered, because they cannot be distinguished from read-through transcripts [69] without information on SVs.

### Permutation test

We performed a permutation test to examine the skew of the distribution of repeat types and strands. Random sequences corresponding to novel exon lengths were extracted from within the gene regions, and overlapping repeat types and their strands were obtained. This procedure was repeated 1,000,000 times, and the distributions of repeat types and their strands were used as a null distribution. The adjustment of the multiple testing was performed using Bonferroni's correction.

### Comparison of gene expressions between HCCs and non-cancerous livers

To compare expression levels in HCCs and matched non-cancerous livers, we used edgeR software [37]. The number of mapped reads for each transcript was normalized using the trimmed mean of M value (TMM) normalization method [40]. Differential expression analysis was performed using the quasi-likelihood method [70]. The Benjamini-Hochberg method was used for multiple testing correction [38].

### Clustering analysis

To compare the expression profiles in HBV- and HCV-infected tissues, we applied k-means clustering to the RNA-seq data. For the log10 converted value of TMM normalized RPM+1, we filtered out candidates with a variance $< 0.1$. We performed k-means clustering and then excluded candidates that were not involved in the clustering by the variance analysis (p-value $< 1 \times 10^{-5}$).

### Cell lines

HepG2 and Huh-7 cell lines were purchased from JCRB Cell Bank. Huh-7 was maintained in Dulbecco's modified Eagle's medium (DMEM; Nacalai tesque), and HepG2 was maintained in DMEM with high glucose (Nacalai tesque). All culture media contained 10% fetal bovine serum (FBS; Gibco) and 50 units/50 μg/mL penicillin-streptomycin (Nacalai tesque).

### Gene overexpression

Cancer-related splicing variants were overexpressed in HepG2 and Huh-7 cell lines (**S11 Table**). The expression vectors were constructed from the pIRES2-AcGFP1-Nuc vector (Clontech). The target sequence was amplified with cloning primers from cDNA (**S11 Table**) and inserted into the EcoR1 site of the pIRES2-AcGFP1-Nuc vector using the In-Fusion HD Cloning Kit (Clontech) according to the manufacturer's protocol. The plasmid was amplified in E. coli DH5α Competent Cells (Takara Bio) and purified using the QIAGEN Plasmid Mini Kit (QIAGEN).

The cells were seeded in 24-well and 96-well plates at a concentration of $1 \times 10^4$/mL from 60% confluent cells. After 48 h of incubation, the expression vectors were transfected into the cells using Lipofectamine 3000 (Invitrogen) according to the manufacturer's protocol. An empty vector of pIRES2-AcGFP1-Nuc was used as control. The medium was changed 2 h after the transfection. For the expression analysis, cells were cultured in 48-well plates, and RNA was extracted 48 h after the transfection. For the cell proliferation assay, cells were cultured in 96-well plates, and cell numbers were measured using the CCK-8 kit (DOJINDO) 96 h after the transfection according to the manufacturer's protocol.

## Gene silencing

To investigate the effect of L1-*MET* on cell proliferation, L1-*MET* expression was suppressed in the HepG2 and Huh-7 cell lines. Cells were seeded in 24-well and 96-well plates at a concentration of $1\times10^5$ cells/mL. After 24 h of incubation, siRNA (**S12 Table**) was transfected into the cells using Lipofectamin RNAiMAX Transfection Reagent (ThermoFIsher SCIENTIFIC) according to the manufacturer's protocol. MISSION siRNA Universal Negative Control #1 (Sigma-Aldrich) was used as a negative control for the siRNA transfection. RNA was extracted after 48 h incubation from the 24-well plates, and the cell proliferation assay was performed for cells in the 96-well plates using CCK-8 (DOJINDO) after 96 h of incubation according to the manufacturer's protocol.

## Supporting information

**S1 Fig. Error patterns in the Nanopore RNA-seq analysis and filtering of SPLICE.**
(PDF)

**S2 Fig. Correlation of expression levels between long-reads and short-reads RNA-seq in MCF-7.** Transcript abundance was measured in reads per million mapped reads (RPM) for long-read RNA-seq data and fragments per kb of exon per million fragments mapped (FPKM) for short-reads RNA-seq data. log10 converted values of RPM+1 and FPKM+1 are shown. r, Pearson's correlation coefficient; p, p-value.
(PDF)

**S3 Fig. Performance of SPLICE pipeline in splicing variant detection.** (A) Number of transcripts detected by SPLICE and four other methods (TALON, FLAIR, StringTie and Bambu). (B) Percentage of transcripts supported in PacBio data with SPLICE and the other methods. (C) Average alignment error rate of SPLICE and the other methods for novel slice sites. The gray dotted line indicates the location of the splice site.
(PDF)

**S4 Fig. Performance of SPLICE pipeline in fusion gene detection.** (A) Venn diagram of the fusion genes detected by SPLICE and two other methods (LongGF and JAFFAL). The neighboring gene pairs (< 200,000 bp) detected by LongGF (MCF-7: VMP1-RPS6KB1, GTF2IRD1-GFT2I and PITPNC1-BPTF, HCC (RK107C): C11orf52-DIXDC1) were excluded from the list because they cannot be distinguished from read-through transcripts without information on structural variations. (B) RT-PCR validation of fusion genes detected by SPLICE, LongGF, and JAFFAL.
(PDF)

**S5 Fig. Percentage of reads mapped to full-length CDS regions.** (A) Schematic diagram of the read length types. "Entire CDS" reads contain all coding exons of known transcripts. "Lacking 3' region of CDS" reads lacks coding exons in the 5' region of known transcripts. "Lacking 5' region of CDS" reads lacks the coding exons in the 3' region of known transcripts. "Lacking both ends of CDS" reads lacks both the coding exon in the 5' region and the 3' region of known transcripts. (B) Pie chart of the detected transcripts by read length type. (C) Relationship between the reference gene length and percentage of read length type in the mapped reads. (Upper panel) Distribution of mapped reads to the reference gene length. (lower panel) Percentage of the mapped read length type to the reference gene length.
(PDF)

**S6 Fig. Correlation between RNA integrity number (RIN) and percentage of "Entire CDS" reads in mapped reads.** r, Pearson's correlation coefficient; p, p-value.
(PDF)

**S7 Fig. Expression levels of long-reads and short-reads RNA-seq in HCC clinical samples.** Transcript abundance was measured in RPM and FPKM for long-reads and short-reads RNA-seq data. log10 converted values for RPM+1 and FPKM+1 are shown. r, Pearson's correlation coefficient. $p < 2.2 \times 10^{-308}$ for all samples.
(PDF)

**S8 Fig. Differential expression levels and structure of AFMID transcripts.** (A) Significantly up- and down- regulated AFMID transcripts in HCCs are shown in red and blue, respectively. (B) Structure of AFMID transcripts.
(PDF)

**S9 Fig. RT-PCR validation of the expression of target transcripts in HepG2 and Huh-7 cell lines.** Target transcript size bands were confirmed by agarose gel electrophoresis. Cell lines transfected with empty vectors were used as control (Ctrl).
(PDF)

**S10 Fig. Effect of L1-MET knockdown on proliferation of HepG2 and Huh-7 cell lines.** (A) Schematic diagram of the target sequence for L1-MET-specific knockdown. (B) Expression levels of L1-MET in L1-MET knockdown in both cell lines.
(PDF)

**S11 Fig. Enrichment analysis of TE-derived exons in HCC-specific transcripts and non-cancerous liver-specific transcripts.** (A) Venn diagram of cancer-specific transcripts and non-cancerous liver specific transcripts. (B) Percentage of TE-derived exons in each item of the Venn diagram in A. P-values for the enrichment of TE-derived exons in cancer-specific transcripts and non-cancerous liver-specific transcripts were calculated by Fisher's exact test. Comparisons were done for TE-derived exons in cancer-specific transcripts and other transcripts and for TE-derived exons in liver-specific transcripts and other transcripts. N.S.: Not significant.
(PDF)

**S12 Fig. Visualization of sequencing coverage in the HBV genome.** HBV transcripts were classified as "HBV (HBV alone)" and "HBV-human genome fusion". (Upper panels) Schematic diagrams of the HBV genome structure. (Lower panels) RNA-seq coverage of HBV transcripts (HBV alone) and HBV-human genome fusion transcripts are shown in blue and red. N.D.: Not detected.
(PDF)

**S13 Fig. Expression analysis of HBV transcripts.** (A) Comparison of HBV transcript expression levels in cancer and matched non-cancerous livers. HBV transcripts were classified as "HBV (HBV alone)" and "HBV-human genome fusion". Total expression levels of transcripts were plotted for each sample. Statistical significances were calculated by the Wilcoxon signed-rank test (B) Expression levels of HBV-human genome fusion transcripts by each transcript type in HCCs and non-cancerous livers. Statistical significances was calculated by the Wilcoxon rank-sum test. (C) Expression levels of HBx-human TE fusion transcripts. HBx-human non-coding fusion transcripts were classified according to the repeat types of the human genome. Their expression levels are shown for each sample. (D) Expression levels of HBx-human TE fusion transcripts in HCCs and livers. Statistical significances was calculated by the

Wilcoxon rank-sum test. (E) Expression levels of HBx-human TE fusion transcripts and non-HBx human TE fusion transcripts in livers. Statistical significances was calculated by the Wilcoxon rank-sum test.
(PDF)

**S14 Fig. Features of fusion transcripts.** (A) Number of mapped bases between long-reads and short-reads RNA-seq. (B) Expression levels of fusion genes detected by both long-reads and short-reads RNA-seq and those detected by long-reads RNA-seq only. P-values were calculated by the Wilcoxon rank-sum test. (C) Average number of exons at the 5' and 3' sides of the fusion gene. (D) Average transcript length (bp) of the 5' and 3' sides of the fusion gene.
(PDF)

**S15 Fig. Expression levels of a novel splicing variant of MYT1L in HCCs and non-cancerous livers.** P-values were calculated by quasi-likelihood methods. The Benjamini-Hochberg method was used for multiple testing correction (FDR).
(PDF)

**S16 Fig. A genome browser view of a novel transcript region of MYT1L identified in this study.** From top, a novel transcript, reference transcripts, and two types of multiple sequence comparisons are shown.
(PDF)

**S17 Fig. Structural and functional analysis of L2-HRH1.** (A) Schematic diagram of the structural differences in L2-HRH1 (NM_001098211.1) and other HRH1 transcripts. (B) Bar chart showing the expression differences between HCCs and non-cancerous livers in HRH1 transcripts. P-values were calculated by quasi-likelihood methods. The Benjamini-Hochberg method was used for multiple testing correction (FDR). (C) Effect of L2-HRH1 overexpression on the proliferation of HepG2 and Huh-7 cell lines as assessed by CCK-8 expression. P-values were calculated by the one-sample t-test (n = 3).
(PDF)

**S18 Fig. Expression levels of epithelial-mesenchymal transition markers (CDH1, JUP and FN1) in HBx-TE fusion overexpressed cell lines.** P-values were calculated by the one-sample t-test ($^{*}p < 0.1$, $^{**}p < 0.05$, $^{***}p < 0.01$).
(PDF)

**S1 Table. Summary of RNA-seq of MCF-7.**
(PDF)

**S2 Table. Primer sequences for validation of fusion genes, related to S3B Fig.**
(PDF)

**S3 Table. Ratios of TE-derived exons among novel exons and known exons.**
(PDF)

**S4 Table. Percentage of BiExp DET-specific and DET and DEG genes.**
(PDF)

**S5 Table. Ratios of TE-derived exons in cancer-specific transcripts.**
(PDF)

**S6 Table. Ratios of TE-derived exons in non-cancerous liver-specific transcripts.**
(PDF)

**S7 Table. Primer sequences for validation of fusion genes, related to Fig 5B.**
(PDF)

**S8 Table. Significantly enriched reactome pathways of genes with up-regulated DETs in HCV-related non-cancerous livers vs. HBV-related non-cancerous livers.**
(PDF)

**S9 Table. Significantly enriched reactome pathways of genes with up-regulated DETs in HCV-related HCCs vs. HBV-related HCCs.**
(PDF)

**S10 Table. Analysis results by SPLICE of sequences basecalled by Guppy v3.0.3 and Guppy v6.0.6.**
(PDF)

**S11 Table. Cloning and RT-PCR primers for overexpression experiments.**
(PDF)

**S12 Table. siRNA design for knockdown experiments and qPCR primers.**
(PDF)

**S1 Data. Clinical and pathological information of the 42 HCC patients.**
(XLSX)

**S2 Data. Summary of RNA-seq of HCC clinical samples.**
(XLSX)

**S3 Data. List of transcripts with novel exons.**
(XLSX)

**S4 Data. List of significantly differentially expressed transcripts.**
(XLSX)

**S5 Data. Significantly enriched reactome pathways of genes with up- or down-regulated DETs.**
(XLSX)

**S6 Data. List of significantly expressed driver gene's transcripts.**
(XLSX)

**S7 Data. List of bi-directionally significantly expressed genes.**
(XLSX)

**S8 Data. List of HBV-human genome fusion transcripts.**
(XLSX)

**S9 Data. List of fusion genes.**
(XLSX)

**S10 Data. Significantly enriched reactome pathways of genes with up-regulated DETs in HBV-infected HCCs vs HBV-infected non-cancerous livers.**
(XLSX)

**S11 Data. Significantly enriched reactome pathways of genes with up-regulated DETs in HCV-infected HCCs vs. HCV-infected non-cancerous livers.**
(XLSX)

## Acknowledgments

The super-computing resource was provided by Human Genome Center, Institute of Medical Science, the University of Tokyo. We acknowledge the staff at Laboratory for Cancer Genomics, RIKEN, for their technical assistances.

## Author Contributions

**Conceptualization:** Akihiro Fujimoto.

**Data curation:** Hiroki Kiyose.

**Formal analysis:** Akihiro Fujimoto.

**Funding acquisition:** Akihiro Fujimoto.

**Methodology:** Hiroki Kiyose.

**Project administration:** Akihiro Fujimoto.

**Resources:** Hidewaki Nakagawa, Atsushi Ono, Hiroshi Aikata, Masaki Ueno, Shinya Hayami, Hiroki Yamaue, Kazuaki Chayama, Akihiro Fujimoto.

**Software:** Hiroki Kiyose, Akihiro Fujimoto.

**Supervision:** Akihiro Fujimoto.

**Validation:** Hiroki Kiyose.

**Writing – original draft:** Hiroki Kiyose, Akihiro Fujimoto.

**Writing – review & editing:** Mihoko Shimada, Jing Hao Wong, Akihiro Fujimoto.

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
