## [Editor Report · Decision Letter 0]

22 Apr 2022

Dear Dr Fujimoto,

Thank you very much for submitting your Research Article entitled 'Comprehensive analysis of full-length transcripts reveal novel splicing abnormalities and oncogenic transcripts in liver cancer' to PLOS Genetics.

The manuscript and Review Commons history was fully evaluated at the editorial level. The reviewers appreciated the attention to an important problem, but raised some substantial concerns about the current manuscript. Based on the reviews, we will not be able to accept this version of the manuscript, but we would be willing to review a much-revised version, as laid out in the included Revision Plan. We cannot, of course, promise publication at that time.

If you decide to revise the manuscript for further consideration at PLOS Genetics, please aim to resubmit within the next 60 days, unless it will take extra time to address the concerns of the reviewers, in which case we would appreciate an expected resubmission date by email to plosgenetics@plos.org.

[LINK]

We are sorry that we cannot be more positive about your manuscript at this stage. Please do not hesitate to contact us if you have any concerns or questions.

Yours sincerely,

Peter McKinnon

Section Editor: Cancer Genetics

PLOS Genetics

---

## [Decision Letter · Decision Letter 1]

12 Jun 2022

Dear Dr Fujimoto,

Thank you very much for submitting your Research Article entitled 'Comprehensive analysis of full-length transcripts reveals novel splicing abnormalities and oncogenic transcripts in liver cancer' to PLOS Genetics.

The manuscript was fully evaluated at the editorial level and by independent peer reviewers. The reviewers appreciated the attention to an important topic but identified some concerns that we ask you address in a revised manuscript

We therefore ask you to modify the manuscript according to the review recommendations. Your revisions should address the specific points made  reviewer 1 & 2.

[LINK]

Yours sincerely,

Peter McKinnon

Section Editor: Cancer Genetics

PLOS Genetics

Reviewer's Responses to Questions

**Comments to the Authors:**

Reviewer #1: I thank the Authors for taking the time to address my concerns.

Regarding their reply, I only need a clarification on their rebuttal to “major point 1”: the Authors state that “If the results are inconsistent (for example, GeneA-GeneB in the reference genome and GeneA-GeneB in the transcriptome genome, or GeneA-GeneB in the reference genome and GeneA in the transcriptome genome), SPLICE considers the candidates as false positive and removes them from the analysis.”. Yet the condition “GeneA-GeneB in the reference genome and GeneA-GeneB in the transcriptome genome” looks consistent to me. This is probably a clerical error since in figure S1 the condition “GeneA-GeneB in the reference genome and GeneA-GeneB in the transcriptome genome” is used to classify the gene-fusion as consistent.

As for the rest, the Authors’ response addressed the points I raised.

I also kindly ask the Authors to provide a web link to access the sequencing data deposited in the National Bioscience Database Center (NBDC). Please clarify where the dataset JGAD000635 has been deposited, is it NBCD (as stated in M&M) or NCBI as stated in the data availability declaration?

Related to this, from the Authors declaration, it is unclear what data will not be made available.

Reviewer #2: Thank you to the authors for your many revisions in response to my and the other reviewers' comments.

1) Perfect, this is great!

2) Also great!

3) This is good.

4) Having the code packaged in Docker certainly helps enormously. I do think that it will be difficult to maintain moving forward.

5) Same re: Docker.

6) Great, I think it's a good result to showcase.

7) This is definitely good to look into. I think you should probably also address recent comments by Oxford Nanopore in talks about how the 109 and earlier sequencing chemistries had issues with mid-transcript priming at homopolymer As, which they claim to have resolved in the 110 and later chemistries. This may be the cause of the issues you observed.

And you should definitely indicate what sequencing chemistry you used and discuss this.

8) Great!

The minor comments have been addressed satisfactorily.

Reviewer #3: The authors have thoroughly and appropriately addressed my concerns from the first round of review.

**Have all data underlying the figures and results presented in the manuscript been provided?**

Reviewer #1: **No: **numerical data and summary statistics are missing

Reviewer #2: Yes

Reviewer #3: None

PLOS authors have the option to publish the peer review history of their article (what does this mean?). If published, this will include your full peer review and any attached files.

Reviewer #1: No

Reviewer #2: **Yes: **Kieran O'Neill

Reviewer #3: No

---

## [Editor Report · Decision Letter 2]

14 Jul 2022

Dear Dr Fujimoto,

We are pleased to inform you that your manuscript entitled "Comprehensive analysis of full-length transcripts reveals novel splicing abnormalities and oncogenic transcripts in liver cancer" has been editorially accepted for publication in PLOS Genetics. Congratulations!

Yours sincerely,

Peter McKinnon

Section Editor: Cancer Genetics

PLOS Genetics

Peter McKinnon

Section Editor: Cancer Genetics

PLOS Genetics

Comments from the reviewers (if applicable):

**Data Deposition**

http://datadryad.org/submit?journalID=pgenetics&manu=PGENETICS-D-22-00465R2

**Press Queries**

---

## [Editor Report · Acceptance letter]

1 Aug 2022

PGENETICS-D-22-00465R2 

Comprehensive analysis of full-length transcripts reveals novel splicing abnormalities and oncogenic transcripts in liver cancer 

Dear Dr Fujimoto, 

We are pleased to inform you that your manuscript entitled "Comprehensive analysis of full-length transcripts reveals novel splicing abnormalities and oncogenic transcripts in liver cancer" has been formally accepted for publication in PLOS Genetics! Your manuscript is now with our production department and you will be notified of the publication date in due course.

With kind regards,

Zsofia Freund

PLOS Genetics

On behalf of:
